# Extended Abstract: A Bayesian approach to phases for frequency-tagged encephalography in the cognitive neuroscience of language

## Abstract

Electroencephalography and magnetoencephalography recordings are non-invasive and temporally precise, making them an invaluable tool in the cognitive neuroscience of language. They are, however, very noisy. One fruitful response to this noisiness has been to use stimuli with a specific frequency and to look for the signal of interest in the response at that frequency. Typically this involves measuring the coherence of response phase. Here a novel Bayesian approach to measuring phase coherence is described and illustrated using an example from neurolinguistics. It gives a better and more data-efficient description than more traditional statistical approaches.

In a frequency-tagged experiment in encephalography stimuli are presented at a specific frequency; for example, in a landmark neurolinguistic experiment Ding et al. (2016, 2017) played four word sentences to participants with each word lasting precisely the same amount of time. The advantage of this approach is that the corresponding magnetoencephalography (MEG) or electroencephalograph (EEG) data can be examined at a few frequencies of interest, in this case, the frequencies which corresponds to words, phrases and sentences. This can give a more robust signal than an ERP-based approach; indeed, there are many situations in neurolinguistics, where an ERP would be impossible to obtain because the number of repetitions required would render the stimulus meaningless to the participant. In Ding et al. (2016) the responses at frequencies corresponding to the phrase and sentence structure of the stimuli demonstrated a neural response which the authors attributed to cortical tracking of hierarchical linguistic structures.

Consider, as an example, the frequency-tagged experiment described in Burroughs et al. (2021a) which follows the frequency-tagged paradigm. This paper investigates the neural response to phrases and compared the response to grammatical adjective-noun (**AN**) phrases such as

  ...cold food loud room...

to ungrammatical adjective-verb (**AV**) pairs, as in

  ...rough give ill tell...

The words are all presented at 3.125 Hz, however the frequency of interest is the 'phrase rate', 1.5625 Hz, corresponding to the phrase structure of the **AN** stimuli. In Burroughs et al. (2021a) it is suggested that comparing the strength of the response for **AN** and **AV** stimuli at the phrase frequency measures a neural response to the grammatical structure, rather than to lexical category, of the words.

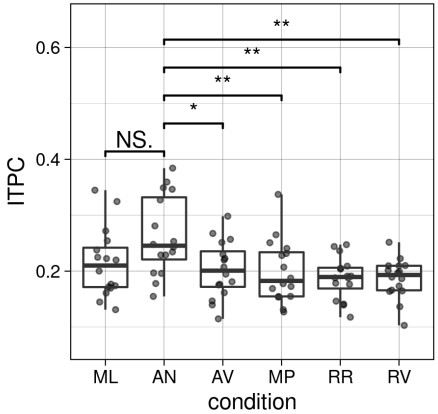

Figure 1: **ITPC results**. The differing response to different conditions is plotted using ITPC; each dot represents a different participants and the horizontal bars indicate statistical differences greater than 0.05 using an uncorrected *ad hoc* pairwise Wilcoxon signed-rank test. This shows all six conditions, the two of particular interest in the paper, **AN** and **AV** and four others, **ML**, **RV**, **MP**, **RR**. The main scientific result of Burroughs et al. (2021a) is the difference between the **AN** and **AV** conditions.

Analysing these data required a quantitative measurement of the strength of the response. The power of the EEG response at 1.5625 Hz does not work, this is too noisy a quantity for the stimulus-dependent signal to be easily detected. Instead the

inter-trial phase coherence (ITPC) was used. The ITPC is defined using the mean resultant

$$r(f, \phi) = \frac{1}{K} \sum_k e^{i\theta_{fk\phi}} \tag{1}$$

where $f$ is the frequency, $k$ is the trial index, $K$ is the number of trials, $\phi$ represents other indices such as electrode number or experimental condition and $\theta_{fk\phi}$ is the phase of the complex Fourier coefficient for the EEG or MEG trace $(f, k, \phi)$. The ITPC is the length of the mean resultant: $|r(f, \phi)|$. In fact, this is averaged over the 32 electrodes used in the experiment to give $|r(f, c)|$ where $c$ indices the conditions. The principle result of Burroughs et al. (2021a) is that, in the language of frequentist statistics, $|r(f = 1.5626 \text{ Hz}, c = \text{AN})|$ is significantly larger than $|r(f = 1.5626 \text{ Hz}, c = \text{AV})|$: Fig. 1.

There are a number of disadvantages to using the ITPC. The most obvious problem is that the 'item' in the statistical analysis of ITPC is a participant, not a trial. In the results described in Burroughs et al. (2021a) the statistical significance relied on a statistical hypothesis test between conditions with a pair of data points for each participant: there are actually 24 trials for each participant but these are used to calculate the ITPC values. This hypothesis testing is performed using 16 pairs of values corresponding to 16 participants, rather than $16 \times 24 = 384$, corresponding to all the trials, or even $16 \times 24 \times 32 = 12288$ items if the individual electrodes are included. In short, the ITPC is itself a summary statistic, a circular version of variance, and so it hides the individual items inside a two stage analysis:

$$\text{items} \rightarrow \text{ITPC} \rightarrow \text{statistical analysis.} \tag{2}$$

Some of the statistical power of the data is lost across these stages. Furthemore, the averaging performed to calculate ITPC ignores much of the structure of the data whereas including, for example, all the electrodes or all the participants equally has the cost of including electrodes less involved in auditory processing and less attentive participants, attempting to select a subset of electrodes or participants is difficult and fraught with statistical risks. These problems are hard to rectify: for example, it is difficult to compare items across participants, or across electrodes, because the mean phase, angle$[r(f, \phi)]$ is very variable and not meaningful to the scientific questions of interest.

Here we introduce a Bayesian analysis to phase data. We believe this has advantages when compared to the ITPC: it permits a per-item analysis and a correspondingly more efficient and richer use of the data. As a Bayesian approach it supports a better description of the data, it can capture information at different levels, participants, trials and electrodes and it replaces a hypothesis-testing and significance-based narrative with a narrative phrased in terms of models and their consequence. The structure of the model is very natural, mimicking the structure of the data and the results are extremely clear, showing a real benefit to the pooling that occurs across items in fitting the model.

In a Bayesian approach a probabilistic model is constructed to account for the data; this is composed of a model of the data and a set of priors for the parameters in that model. For the phase data considered here the data is a set of angles and so the model for the data is a probability distribution on a circle. The motivation which informs the ITPC is that the angles to a greater or lesser extent have a unimodal distribution. To match this assumption, the model used here should be a unimodal distribution on the circle. One common class of distributions on the circle is given by the so called wrapped distributions with probability density function $p(\theta) = \sum_{n=-\infty}^{\infty} p_r(\theta + 2\pi n)$ where $p_r(x)$ is the probability density of a distribution on the real line. Here the Cauchy distribution is used, remarkably, the wrapped Cauchy distribution has a closed form:

$$p(\theta) = \frac{1}{2\pi} \frac{\sinh \gamma}{\cosh \gamma - \cos(\theta - \mu)} \tag{3}$$

A large value of $\gamma$ corresponds to a highly dispersed distribution; a low value to a concentrated one, see Fig 2**A**. $\mu$ is the location of the peak.

The next important element is the choice of priors both for $\mu$ and for $\gamma$. In a sense the prior for $\mu$ is more straight-forward: a value of $\mu$ is required for each electrode and for each participant. Here for simplicity an independent value $\mu_{epc}$ is chosen for each triplet of electrode-participant-condition values. There is also no preferred value for $\mu$: the mean phase depends on processing and transmission latencies and their relationship to the frequency, see Fig. 2**C**. As such the correct prior is uniform on a circle. In principle there is simple; nothing in the mathematical description of the common MCMC samplers, such as Metropolis-Hastings, HMC or NUTS, prevent the prior from

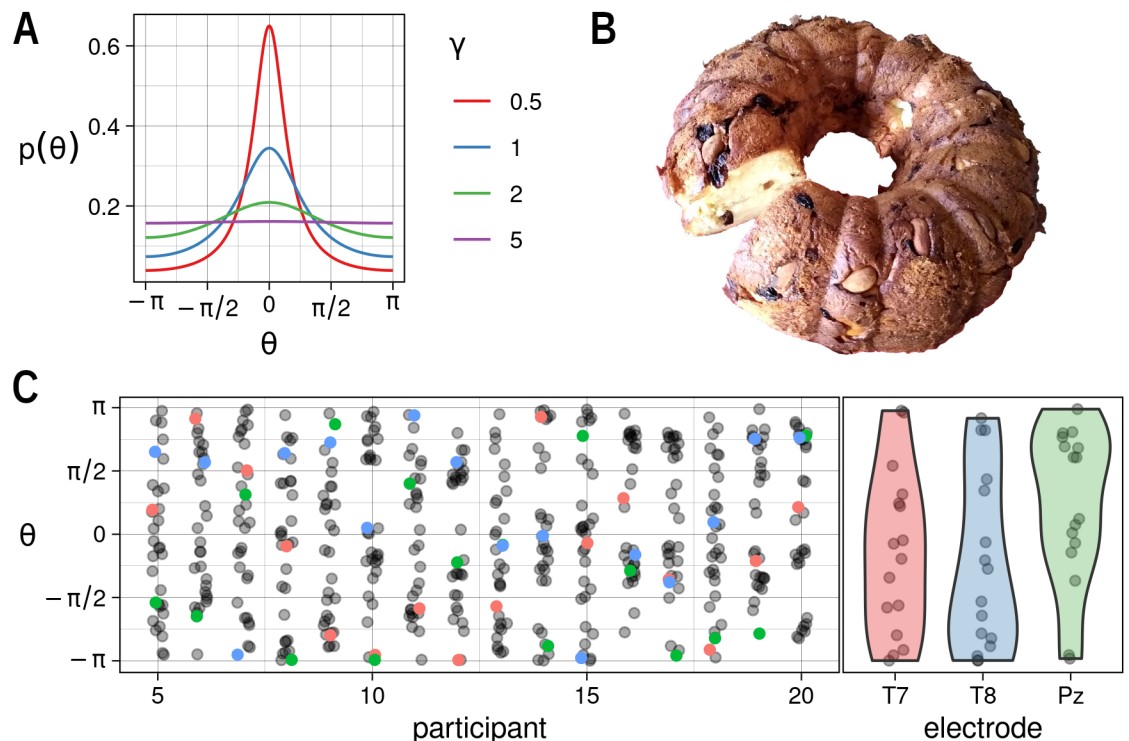

Figure 2: **Circular distributions**. The wrapped Cauchy distribution for $\mu = 0$ and for four different values of $\gamma$ is plotted in **A**; the $\gamma = 0.5$ graph is rather peaked and corresponds to a mean resultant with $|r| \approx 0.61$, in contrast $\gamma = 5.0$ is very flat and corresponds to $|r| \approx 0.01$. The bundt distribution has a shape reminiscent of a cake, like the one in **B** made in a bundt tin. To demonstrate the suitability of a uniform circular prior for the mean angle, the direction of the mean resultant for all sixteen participants is plotted in **C**, where all 32 electrodes are plotted for condition AN at the phrase frequency. To see how a specific electrode varies across participant three have been colored, T7 in red, T8 in green and Pz in blue, the violin plots corresponding to these electrodes highlight the uniform spread of the mean resultant around the circle.

being defined on a compact region. However, there is a problem in practice: the current high-quality implementations of these methods in `stan` and `turing.jl` do not allow priors of this sort.

As a practical approach to avoiding this difficulty, we introduce a two-dimensional distribution which, in polar coordinates, is uniform in the angle coordinate. That angle coordinate is used as the mean angle in the wrapped-Cauchy distribution, giving the uniform prior on a circle this variable requires. Because its probability density function resembles the bundt cake tin, used to make kugelhopf, this will be referred to as a bundt distribution, see Fig. 2**B**. It has probability density function

$$p(\mathbf{x}) = \frac{1}{2\pi} \rho \exp\left(-\rho\right) \quad (4)$$

where $\rho = |\mathbf{x}|$. The idea here is to have a prior

$$\mathbf{x}_{epc} \sim \text{Bundt}() \quad (5)$$

and then set

$$\mu_{epc} = \text{angle}(\mathbf{x}_{epc}) \quad (6)$$

The final element of the model is the prior for $\gamma$; obviously the intention is have this depend on the condition. One simple model uses a log-logistic link function:

$$
\begin{aligned}
\alpha_c, \alpha_e, \alpha_p, \alpha &\sim \text{Normal}(0, 1) \\
s &\sim \text{Exponential}(1) \quad (7)
\end{aligned}
$$

where $e$ is the electrode index, $p$ the participant index, $c$ the condition index; different $\alpha$s with different indices are intended to be understood as different variables. Next

$$r(c, e, p) = \ell[s^2(\alpha_c + \alpha_e + \alpha_p + \alpha)] \quad (8)$$

where $\ell(x)$ is the logistic function and

$$\gamma_i = -\log r(c[i], e[i], p[i]) \quad (9)$$

where $i$ is the item index for the data. The likelihood for the angles is then

$$\theta_i \sim \text{WrappedCauchy}(\mu_{e[i]p[i]c[i]}, \gamma_i) \quad (10)$$

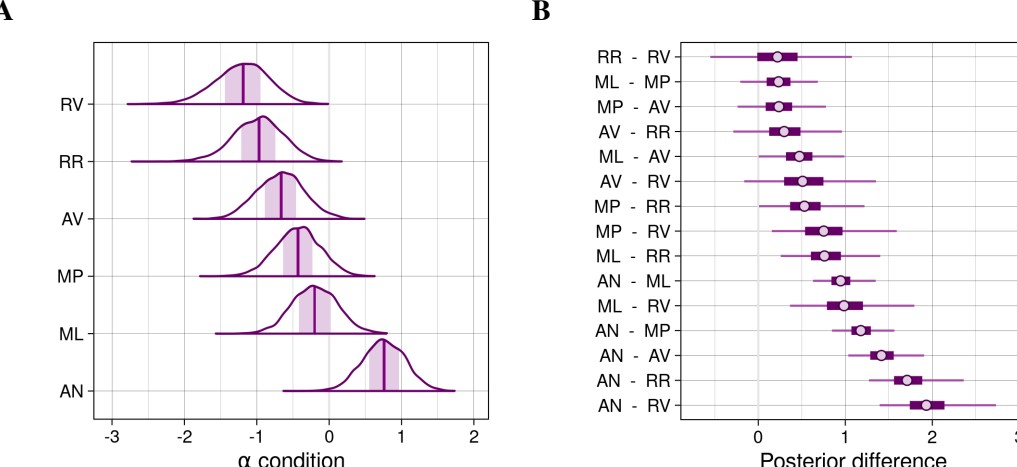

Figure 3: **Posterior distributions**. The marginal density plots in **A** show the posterior distributions of $\alpha_c$. The posterior distribution shows a clear progression with the **AN** condition showing the strongest response to the stimulus and the **RV**, a random control, the weakest. The highest probability density intervals in **B** show the progression in the strength of response from stimulus to stimulus. For the posterior distribution of difference between conditions, $\alpha_{c_1} - \alpha_{c_2}$, the circle is the mean and the whiskers span the shortest interval which includes 97% of the probability density.

where $\theta_i$ is the $i$th phase.

The data used here was previously described in Burroughs et al. (2021a) where there is a detailed description of the experimental procedures; the data are freely available at Burroughs et al. (2021b). The posterior distributions were sampled using the probabilistic programming language `stan` and are illustrated in Fig. 3. This gives an extremely clear set of results; whereas the original frequentist analysis used in (Burroughs et al., 2021a), Fig. 1, only showed that the stimulus response for the **AN** condition was higher than other conditions, the Bayesian approach described here tells a richer and more more complete story. As an example, from Fig. 3**B** it is clear that there is a difference in the response in the model for the **ML** condition in comparison to **RR**: this is interesting since the **ML** condition is not a grammatical condition. Further analysis, beyond what is shown here, shows the Bayesian result is more robust as participants are removed from the analysis, showing greater descriptive power.

A Bayesian analysis is very straightforward: it provides samples of the posterior distributions for model parameters based on data, these can be used to test hypothesis phrased in terms of a model. Here we have outlined how a Bayesian model can be constructed for the phase data produced by frequency tagging experiments and we are preparing a full description of the mathematical background, of different Bayesian models, their robustness, the code, libraries and packages used to analyze them and the results they produce. It is clear that the results provide an efficient approach to these data, an important type of data for the cognitive neuroscience of language.

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
