# OpenReview forum: "Extended Abstract: A Bayesian approach to phases for frequency-tagged encephalography in the cognitive neuroscience of language"
_aclweb.org/ACL/2022/Workshop/CMCL — CMCL 2022 nonarchival_

### Official Review · Reviewer_Pgx7 · 2022-03-16
**a nice methodological note regarding a previous analysis**

**Rating:** 7
**Confidence:** 3

**Review:**

This would make a nice poster at CMCL. It simply lays out a Bayesian version of Frequentist analysis that was used in a published paper last year. Although there is no cognitive modeling per se, the method itself will be interesting to a variety of CMCL attendee. There is probably *not* enough here for an article in a journal like _Bayesian Analysis_ so it's appropriate that we offer a venue in which to share this work.

---

### Official Review · Reviewer_vD5a · 2022-03-23
**A Bayesian analysis of a neurolinguistic EEG study**

**Rating:** 7
**Confidence:** 3

**Review:**

This submission describes a Bayesian analysis of frequency-tagged EEG experiments, a type of experimental paradigm used in neurolinguistics research. The analysis is meant to serve as an alternative to a traditional frequentist analysis from prior works, which averages over electrodes and items instead of explicitly accounting for associated variance.

Main contributions of the work:
1.	A discussion of a Bayesian EEG analysis as an alternative to frequentist methods
2.	An attempt to explicitly model item and participant effects
3.	A nuanced discussion of Bayesian priors required to model inter-trial phase coherence (ITPC) effects

Questions and clarifications:
1.	The authors pitch the need to model item and participant effects as the main incentive for switching to Bayesian analysis. However, there is a frequentist approach that achieves the same effect — mixed effects modeling. Mixed effects models have been gaining popularity for EEG analyses, including analyses of ITPC (see e.g. Koerner & Zhang, 2017). Of course, this doesn’t mean that the Bayesian analysis has no value, but I would recommend the authors to adjust their motivating arguments (and possibly mention mixed effects models as a frequentist alternative)
2.	Differential effects across electrodes/participants. The authors correctly state that averaging the data across electrodes and participants “washes out” effects of interest that vary across them. However, their analysis doesn’t quite seem to address this issue: the effect of condition is modeled to be the same across all electrodes and participants if I understand it correctly. Thus, while modeling the variance in participant-, electrode-, and condition-specific averages is certainly a step forward, differential effects across participants and electrodes remain unaccounted for in this model.
3.	How much data are required for the proposed analysis? It might be helpful to have a brief discussion of guidelines for determining whether the number of, e.g., item repetitions across participants is sufficient to fit the model.
4.	In eqs. 7 and 8, does alpha without a subscript correspond to the baseline value shared across participants, electrodes and conditions? The current wording is a bit ambiguous on that point.
5.	The justification for using the log-logistic link function to model gamma is unclear to me (an outsider to this subfield). We don’t need to know the specifics but a one-line explanation or a reference would help.


Minor:
-	To better advertise the analysis for scientists used to frequentist methods, it might be useful to include Bayesian factor as a more direct alternative to frequentist hypothesis testing
-	In Figure 3, you might want to highlight the key conditions of interest

Overall, this is an interesting contribution to the field of EEG data analysis, with clear applications in the domain of neurolinguistics.

---

### Official Review · Reviewer_khn9 · 2022-03-27
**A Bayesian model of phase coherence in frequency-tagged EEG data**

**Rating:** 7
**Confidence:** 2

**Review:**

This paper discusses disadvantages of a frequentist ITPC analysis (such as loss of statistical power, no full characterization of the structure of the data) of frequency-tagged EEG data, and introduces a Bayesian model as a novel alternative.
The model includes:
* a wrapped Cauchy distribution as a model of the data (a set of angles drawn from a unimodal probability distribution on the circle)
* a prior for the mean phase for each participant-electrode-condition triplet as a uniform on a circle
* a condition-dependent prior for the scale parameter of the distribution, resulting in different mean resultants

The authors show that the posterior distributions yielded by this model give a richer and more complete picture of the data than the original frequentist ITPC analysis.

**General**

The paper is clearly written and of theoretical interest to the community. The characterization of the data derived via the analysis allows for interesting new conclusions that were not detectable using the ITPC method, which are relevant to the original research question. As someone without a background in EEG time-series analysis, the proposed Bayesian model nevertheless strikes me as straightforward and well-motivated. To make the novel contribution of the paper even stronger I suggest a discussion of how this approach compares to a mixed-effects model, which can also model the random effects in the data that ITPC as a summary statistic glosses over.

**Specific questions/Comments**
* One thing that wasn’t clear to me from the write-up is how the simplifying assumption of choosing independent location parameters per participant-electrode-condition triplet instead of per participant-electrode pair affects the model results (lines 144ff).
* I would have liked to see the unabbreviated names for all 6 conditions somewhere in the paper, especially since the ML condition for which the model provides a novel result in relation to RR is not listed as an experimental condition in the Burroughs et al. (2021) paper; The same is true for the RV condition, but this name is more transparent given the other conditions. I assume ML and RV correspond to the two extra conditions described as fillers in the Methods section of Burroughs et al. (2021), and analyzed in Figure 2 of their Supplementary Materials, where RV = RRRV and ML = ADVP, but it would be good to make this correspondence clear. Furthermore, if this alignment is correct it may be good to discuss a potential effect of these two streams having 12 four-word phrases instead of 24 two-word phrases as the other conditions on the critical ML-RR modeling results.

Overall, I think this paper provides an interesting contribution and I am looking forward to seeing the full description of the different Bayesian models and their robustness advertised at the end of the paper.

---

### Decision · Program_Chairs · 2022-03-29

Accept (non-archival)